# Global spread of *Salmonella* Enteritidis via centralized sourcing and international trade of poultry breeding stocks

Shaoting Li[1,2], Yingshu He[1,2], David Ames Mann[1] & Xiangyu Deng 📧 [1✉]

A pandemic of *Salmonella enterica* serotype Enteritidis emerged in the 1980s due to contaminated poultry products. How *Salmonella* Enteritidis rapidly swept through continents remains a historical puzzle as the pathogen continues to cause outbreaks and poultry supply becomes globalized. We hypothesize that international trade of infected breeding stocks causes global spread of the pathogen. By integrating over 30,000 *Salmonella* Enteritidis genomes from 98 countries during 1949–2020 and international trade of live poultry from the 1980s to the late 2010s, we present multifaceted evidence that converges on a high likelihood, global scale, and extended protraction of *Salmonella* Enteritidis dissemination via centralized sourcing and international trade of breeding stocks. We discovered recent, genetically near-identical isolates from domestically raised poultry in North and South America. We obtained phylodynamic characteristics of global *Salmonella* Enteritidis populations that lend spatiotemporal support for its dispersal from centralized origins during the pandemic. We identified concordant patterns of international trade of breeding stocks and quantitatively established a driving role of the trade in the geographic dispersal of *Salmonella* Enteritidis, suggesting that the centralized origins were infected breeding stocks. Here we demonstrate the value of integrative and hypothesis-driven data mining in unravelling otherwise difficult-to-probe pathogen dissemination from hidden origins.

---

[1] Center for Food Safety, University of Georgia, Griffin, GA, USA. [2] These authors contributed equally: Shaoting Li, Yingshu He. ✉email: xdeng@uga.edu

Poultry products including meat and eggs are among the most important sources of dietary protein for human consumption. In 2018, chicken became the predominant meat in the world as its projected global production surpassed that of pork[1]. In 2018, there were over 29 billion poultry in the world[2]—roughly three birds for each person on the planet, and about five times >50 years ago[3].

Industrialized poultry production features a breeding pyramid with genetic stock situated at the top and supplying the entire industry (Fig. 1). Breeding stocks, mostly day-old grandparent and parent stock chicks, are provided to production farms to produce meat and eggs. Through the poultry production hierarchy, the sizes of poultry flocks are massively amplified[4] (Fig. 1).

Genetic improvement of commercial poultry stocks began in the 1940s. In 1948 and 1951, the national "Chicken of Tomorrow" contests were held in the United States to challenge breeders to find or develop a superior meat-type bird[5]. Several breeders emerged from the contests and became established market brands[6]. The emergence of breeding selection and specialized breeders heralded structural transformation and consolidation of the poultry industry, which evolved into one of the most integrated agribusinesses as the major poultry markets in the US and Europe matured in the 1980s and the early 1990s[6,7].

Driven by the capital and technology intensive nature of pedigree breeding, significant merging and acquisition of breeds or brands took place since the early 1980s[8]. As a result, most commercial broiler chickens worldwide originate from nucleus breeding flocks[6,9]. A similar structure and concentration exists in the supply of layer breeding stocks[10].

While providing the world population with a significant source of animal protein, the success of industrialized poultry farming has been accompanied by the spread of Salmonella enterica, one of the most common causes of foodborne illness. In the 1980s, simultaneous increase of Salmonella enterica serotype Enteritidis infections linked to poultry occurred in North America, South America, and Europe[11]. By the late 1980s and the 1990s, Salmonella Enteritidis had spread to the poultry production systems in Asia[12] and Africa[13]. The pandemic subsequently declined in the late 1990s in the US[14] and the United Kingdom (UK)[15], but Salmonella Enteritidis remains a substantial problem for poultry production and public health. Between 2015 and 2018, Salmonella Enteritidis-contaminated eggs caused the largest known salmonellosis outbreak in Europe, resulting 1,209 reported cases in 16 countries[16].

Decades after the inception of the pandemic, how the pathogen simultaneously increased in many parts of the world remained "a central and unsolved mystery"[17]. We noticed a spatiotemporal coincidence between the onset of the Salmonella Enteritidis pandemic and the beginning of global breeding stock consolidation, both of which first occurred in America and Europe in the 1980s and subsequently affected other continents. We therefore hypothesize that centralized sourcing and international trade of Salmonella Enteritidis-infected breeding stocks is a parsimonious explanation for the synchronized and expansive spread of Salmonella Enteritidis.

Circumstantial or anecdotal evidence from different parts of the world supports the hypothesis. Salmonella Enteritidis or other S. enterica serotypes were detected from primary breeding flocks in the UK in 1988[18] and in the US in 1999[19], two of the earliest and major exporters of breeding stocks. Salmonella control in imported grandparent stocks was speculated to spare Sweden from the Salmonella Enteritidis pandemic[20] and clear Salmonella Enteritidis infections from Netherlands' national flock[21]. In Brazil, Salmonella Enteritidis was isolated from transport boxes of imported day-old chicks between 1997 and 1998[22]. Subtyping of a national collection of isolates from 1986 to 2010 suggests a common origin of circulating Salmonella Enteritidis strains in the country[23]. In Japan, Salmonella Enteritidis with a potential to cause transovarial infection to progeny birds was detected in chicks imported from England in 1988 and 1989[24]. Japan saw an increase of Salmonella Enteritidis cases and a replacement of the previous dominant Salmonella Enteritidis clone by a new strain since 1989[12]. In Egypt, Salmonella Enteritidis was isolated from imported day-old chicks in the early 2000s[25]. The country imported over 110 million live birds in 2006[25].

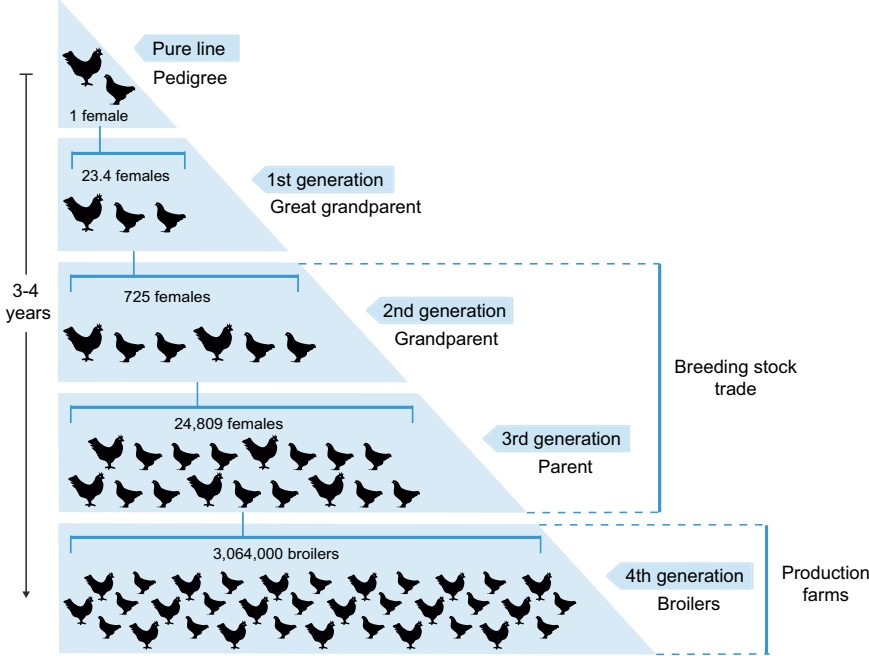

**Fig. 1 Breeding pyramid of industrialized poultry production.** Poultry flock sizes derived from a single pedigree female chicken was according to the estimate by Van Eenennaam et al.[4].

Despite these reports and speculations, the chain of evidence to link the global spread of *Salmonella* Enteritidis to the international trade of breeding stocks is incomplete.

First, there is no evidence that suggests that a foreign strain of *Salmonella* Enteritidis can traverse the domestic poultry production hierarchy of a breeding stock importer (Fig. 1). This is a pre-requisite for imported *Salmonella* Enteritidis to cause widespread and sustained contamination in production flocks, which may ultimately lead to human epidemics.

Second, the spatiotemporal scale of the possible *Salmonella* Enteritidis spread via breeding stocks has not been evaluated to determine whether this mode of transmission is adequate to cause a pandemic and may still be underway. Such an evaluation is important for solving the historical puzzle of the *Salmonella* Enteritidis pandemic, and will be valuable for prospective monitoring and intervention of emerging pathogens in poultry. Poultry production is a notable venue for repeated emergence additional *S. enterica* serotypes, including serotypes Pullorum and Gallinarum, Enteritidis, Heidelberg, and Kentucky[26]. Most recently, closely-related, multidrug-resistant strains of serotype Infantis have spread through poultry in Europe[27] and the US[28]. It is unclear whether a common anthropogenic factor catalyzed the multiple waves of surge and spread of *Salmonella* in poultry.

Filling the gaps is challenging because public surveillance of *Salmonella* in breeding stocks is sparse and independent data to detail trade flows by individual providers do not exist[9]. Strong emphasis on intellectual property protection[29] and high standards of biosecurity[9] make primary breeding less transparent to public scrutiny[30]. However, whole genome sequencing (WGS) of *Salmonella* Enteritidis from poultry products and production environments is burgeoning for global surveillance of *Salmonella*. International trade of live chickens including breeding stocks is well documented for tariff reporting. Both types of data are abundant and publicly available.

To test our hypothesis, we performed a genomic epidemiology study by integrating global genomic surveillance of *Salmonella* Enteritidis and international trade of live poultry. Here we present multifaceted evidence to inform the likelihood, scale, and protraction of *Salmonella* Enteritidis spread through centralized sourcing and international trade of breeding stocks.

## Results

**Salmonella Enteritidis isolates recovered from poultry in different continents are genetically related.** We analyzed WGS data of 30,015 *Salmonella* Enteritidis strains isolated from 98 countries between 1949 and 2020 that were available at EnteroBase[31] as of November 2020. We identified multiple cases of genetically related poultry isolates collected from different continents that were separated by a maximum of five alleles by core genome multi-locus sequence typing (cgMLST). Six isolates sampled from postharvest or retail chicken in five US states between 2015 and 2020 differed by one cgMLST alleles or 0–7 single-nucleotide polymorphisms (SNPs) from an isolate detected in a pre-harvest broiler bird in Suriname in 2016 (Fig. 2 and Supplementary Data 1). Such a high level of genetic closeness is typically observed among isolates from a single monoclonal outbreak[32], suggesting a common source of contamination. Import of breeding stocks to Suriname, including both live birds and fertilized eggs, predominately originated from the US from 2010 to 2016 (Supplementary Fig. 2).

Other countries linked by possible intercontinental transmission of *Salmonella* Enteritidis via poultry include Netherlands-Brazil, Denmark-Brazil, Turkey-China, and Colombia-USA (Supplementary Data 1). In these cases, isolates from the linked countries were

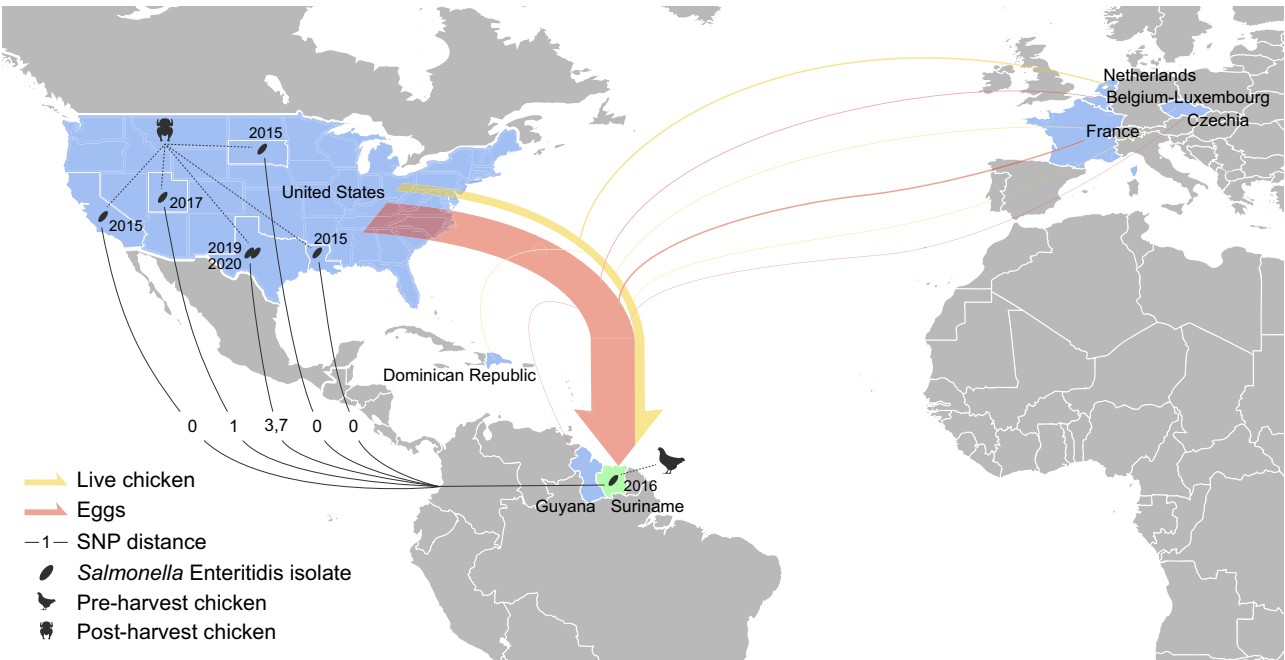

**Fig. 2 Closely related poultry *Salmonella* Enteritidis isolates from Suriname and US and breeding stocks export to Suriname.** Arrows indicate the export of live chickens (yellow) and eggs (red) to Suriname. Export data between 2010 and 2016 from the US and other countries were obtained from the US Foreign Agricultural Service (FAS) and the Observatory of Economic Complexity (OEC), respectively. The sizes of arrows are proportional to the monetary values of export. Export of live chicken from the US includes meat-type and egg-type breeding stocks specifically, while that from the other countries includes day-old chicks <185 g. Export of eggs from the US includes only fertilized eggs, while that from the other countries includes shelled, fresh, preserved or cooked eggs. *Salmonella* Enteritidis isolates are indicated by ovals at the locations of their isolation along with the isolation years. The Suriname isolate was detected in a pre-harvest broiler bird. The US isolates were detected from post-harvest chickens. The numbers on the solid lines indicate pairwise core genome SNP distance between the Suriname isolate and each US isolate.

collected from 2006 to 2018 and separated by 1–4 cgMLST alleles. It is unknown whether these isolates were collected from domestically raised chicken or processed poultry products of foreign origin. Trade of both breeding stocks and processed poultry was identified between the linked countries prior to the discovery of the isolates (Supplementary Fig. 1), suggesting the possibility of either or both commodities as a vehicle for transmission of *Salmonella* Enteritidis. We also observed eight instances where a human isolate from one continent was closely related to a poultry isolate from another continent (Supplementary Data 1). These observations may indicate *Salmonella* Enteritidis spread via poultry, but travel-related transmission cannot be excluded.

**Isolates of additional serotypes support intercontinental transmission via poultry.** In addition to *Salmonella* Enteritidis, we found pre-harvest chicken isolates of serotype Ohio in Suriname ($n = 1$) and of serotype Kentucky in Barbados ($n = 1$) and Trinidad and Tobago ($n = 1$) that were closely related to poultry isolates in the US (1 to 5 cgMLST alleles) (Supplementary Data 1). The Caribbean and US isolates were sampled from 2016 to 2019. Both Caribbean countries imported breeding stocks predominantly from the US from 2010 to 2016 (Supplementary Fig. 2).

**Isolates from poultry are more closely related than isolates from other sources.** We investigated the population structure of the 30,015 *Salmonella* Enteritidis isolates using cgMLST[31]. The isolates were obtained from 32 sources including poultry[31]. Poultry isolates were from 28 countries and displayed tight clustering in two major lineages (Fig. 3a).

Eight sources each contained isolates from at least 100 cgMLST sequence types (cgSTs). We assessed genomic diversity of isolates within each source by calculating all pairwise allelic differences (PADs) between two isolates from the source. Poultry isolates showed the smallest intra-source genomic diversity with a median PAD of 110 compared with isolates from other sources whose median PADs ranged from 152 to 357 (Fig. 3b).

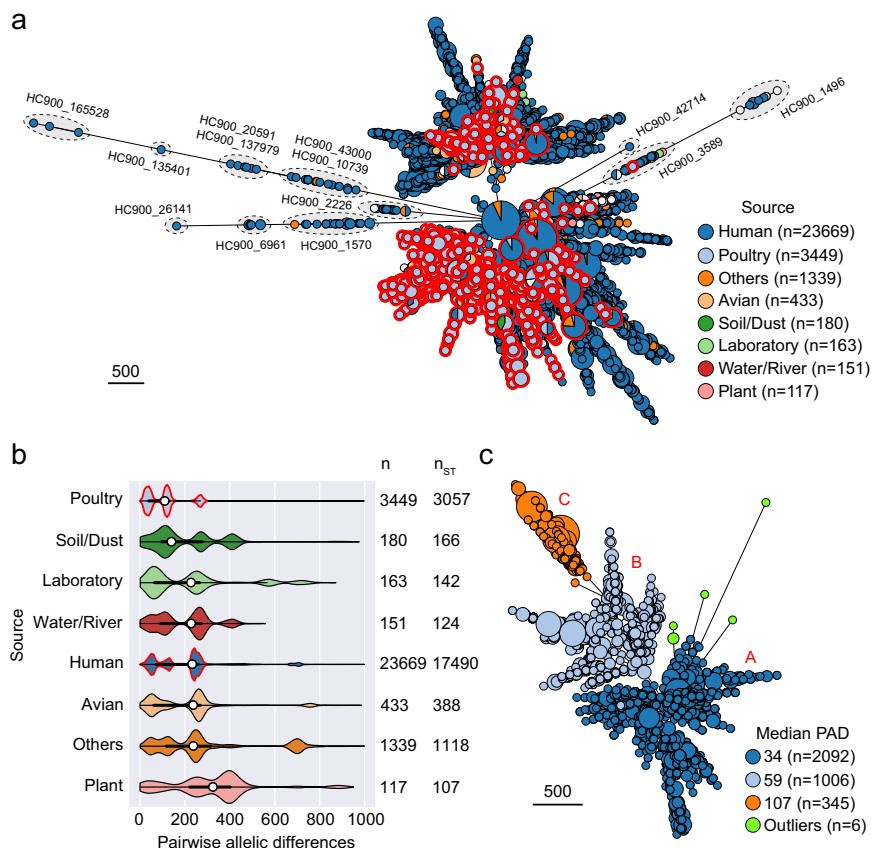

**Fig. 3 Global population structure and genomic diversity of *Salmonella* Enteritidis. a** Minimum spanning tree of 30,015 *Salmonella* Enteritidis isolates of 22,775 cgMLST sequence types (cgSTs) in EnteroBase. Each node represents a distinct cgST or cgSTs differed by only missing data. The size of each node indicates the number of isolates within that node. Isolates are color coded by EnteroBase-defined sources. Only sources with at least 100 cgSTs are shown. Nodes containing poultry isolates are highlighted by red border. The majority of isolates belong to the HC900_12 cluster defined by EnteroBase. Clusters not belonging to HC900_12 are highlighted by dashed ovals. The scale bar indicates 500 cgMLST alleles. An interactive version of the tree is available at https://enterobase.warwick.ac.uk/ms_tree/51520. **b** Comparison of intra-source pairwise allelic differences (PADs) based on individual STs. Violin plots show probability densities of PADs. Each boxplot within a violin plot summarizes the distribution of PADs of all isolates in a source. White circles indicate the median PAD values. Thick black bars represent the interquartile ranges. Thin black lines indicate the ranges between maximum and minimum values. Only sources with numbers of isolates ($n$) and cgSTs ($n_{ST}$) $\geq 100$ are shown. "Others" includes isolates without source information or not belonging to any specific source defined in EnteroBase. PADs larger than 1,000 are considered outliers and not included. The peaks of kernel density estimate of the PAD distributions of poultry and human isolates are outlined in red. **c** Minimum spanning tree of 3,449 *Salmonella* Enteritidis poultry isolates of 3,057 cgSTs in EnteroBase. Three sub-populations of poultry isolates (A, B, and C) and their individual median PADs are shown. Six outlier isolates (green) belong to four different HC200 clusters including HC200_13575 ($n = 3$), HC200_12160 ($n = 1$), HC200_1522 ($n = 1$). The rest of isolates in A, B, and C belong to the same HC200 cluster, HC200_12. The scale bar indicates 500 cgMLST alleles. An interactive version is available at https://enterobase.warwick.ac.uk/ms_tree/50727.

We observed a multimodal distribution of PADs in every source (Fig. 3b), indicating heterogeneous populations of *Salmonella* Enteritidis commonly circulating in such sources. Using cgMLST, we identified three such populations of poultry isolates whose individual median PAD was 34, 59, and 107 (Fig. 3c). Poultry and human isolates displayed parallel PAD distributions with their respective peaks of kernel density estimate occurring at similar positions (Fig. 3b).

**Phylodynamics of poultry *Salmonella* Enteritidis lineages lends spatiotemporal support for their dispersal from centralized origins during the pandemic.** To probe the spatiotemporal spread and evolutionary origins of circulating *Salmonella* Enteritidis lineages in poultry, we performed phylodynamic analyses on a representative set of 914 *Salmonella* Enteritidis genomes from 46 countries between 1954 and 2020. These genomes were selected to represent the global diversity in poultry *Salmonella* Enteritidis phylogeny and epidemiology, avoid redundant sampling of similar isolates, and provide a broader phylogenetic context by including isolates from non-poultry sources such as humans (see "Methods").

From the global SNP phylogeny, we identified three major lineages of poultry isolates: an international lineage represented by isolates from all inhabited continents (Global); another international lineage dominated by European and US isolates (Atlantic), including those from the largest European outbreak of salmonellosis in 2015–2018[16]; and an overwhelmingly US lineage (US), which has the aforementioned Suriname isolate (Fig. 4a). The Atlantic and the US lineages are more closely related to each other than either is to the Global lineage. Notably, the Global and the Atlantic lineages each correspond to a distinct poultry *Salmonella* Enteritidis population delineated by the measurement of intra-population genomic diversity (Global: "C" and Atlantic: "B" in Fig. 3c), while the US lineage scatters in two such populations ("A" and "B" in Fig. 3c). Human isolates included in this analysis intermingle with

poultry isolates in all three lineages (Fig. 4a), which is consistent with the aforementioned parallel distribution of PADs between poultry and human isolates (Fig. 3b). For eight isolates representing all three major lineages, their phage types have been pre-determined and published. These isolates and their phage types were denoted in Fig. 4a and Supplementary Data 2.

Within poultry, the mixture of isolates from chicken and eggs suggests indistinguishable *Salmonella* Enteritidis populations circulating in both broilers and layers (Fig. 4a). For each egg isolate on the phylogeny, its closest chicken isolate was 0 to 125 cgMLST alleles (median: 17) or 0 to 101 SNPs (median: 15) away. For chicken and egg isolates in the Global lineage that exhibited strong temporal signals for robust ancestral state inference, it was common to observe egg isolates with an inferred chicken ancestor and vice versa, oftentimes with such ancestors and their progenies separated by <5 years (Supplementary Fig. 3).

The Global lineage contains poultry *Salmonella* Enteritidis clades specific to individual countries, such as Chile and Brazil (Fig. 4a). Relatively long root-to-tip distances of these clades and unresolved branching order among their ancestral nodes suggest clonal expansion and diversification of domestic populations that are descended from closely related progenitors (Fig. 4a). In these countries, the early pandemic strains may have persisted and become endemic. By contrast, a cluster of poultry isolates from Mauritius are apically positioned in the Atlantic lineage with considerably less diversification, suggesting relatively recent introduction. Similarly, the Suriname isolate is found in the US lineage amid isolates characteristic of recent clonal expansion.

We performed an exhaustive search for temporal signals of SNP accumulation by screening every internal node of the phylogeny (i.e., inferred common ancestors) for strong correlation between isolation years of and branch lengths to the corresponding tips (i.e., extant isolates) of the phylogeny. The Global and the Atlantic lineages exhibited strong such signals ($R^2 > 0.4$, Fig. 4b), which allowed robust and evolutionary modeling-based inferences of clade ages,

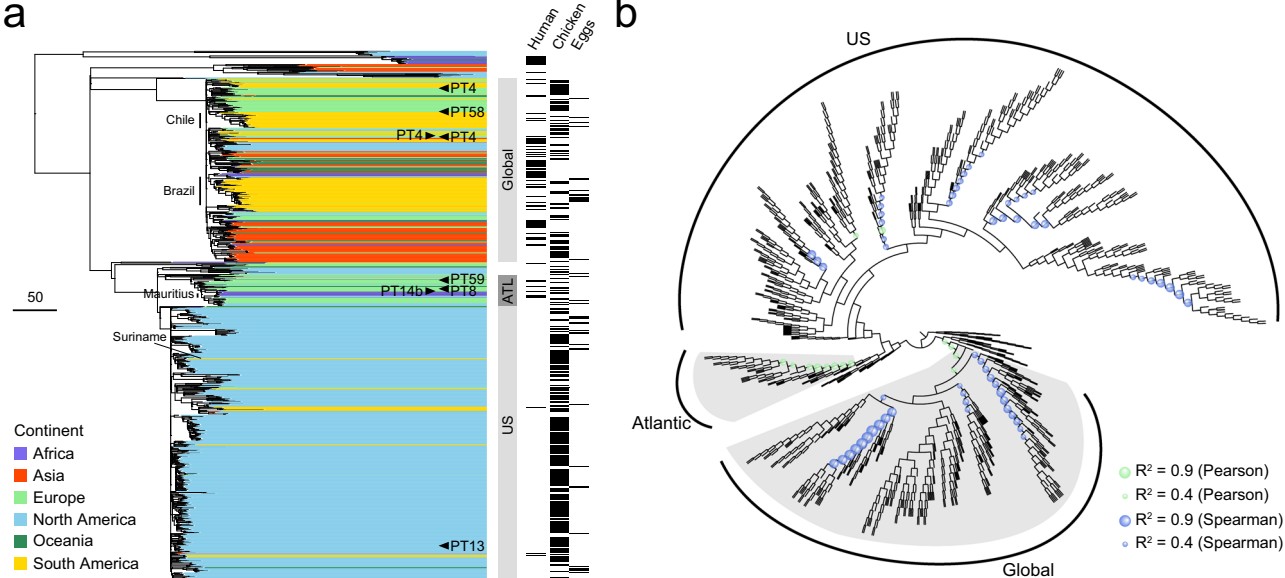

**Fig. 4 Global phylogeny *Salmonella* Enteritidis from poultry. a** Maximum-likelihood phylogeny of 914 selected poultry, human, and other isolates from 46 countries. The tree is rooted at midpoint. Three major lineages (Global, Atlantic, and US) are delineated. Each isolate is color coded by the continent of origin. The isolates from humans and poultry (including chickens and eggs) are indicated. Poultry *Salmonella* Enteritidis clades specific to Chile, Brazil, and Mauritius are delineated. Arrowheads indicate isolates whose phage types are publicly available. ATL, Atlantic. The scale bar indicates 50 SNPs. **b** Circular cladogram of the same maximum-likelihood phylogeny of the 914 isolates. Colored circles indicate internal nodes that have a squared coefficient ($R^2$) of the Spearman or Pearson correlation between isolation years and branch lengths >0.4. The sizes of the circle are proportional to the values of $R^2$ (0.4–0.9). The Global and the Atlantic lineages that exhibit strong temporal signals of SNP accumulation are shaded in gray.

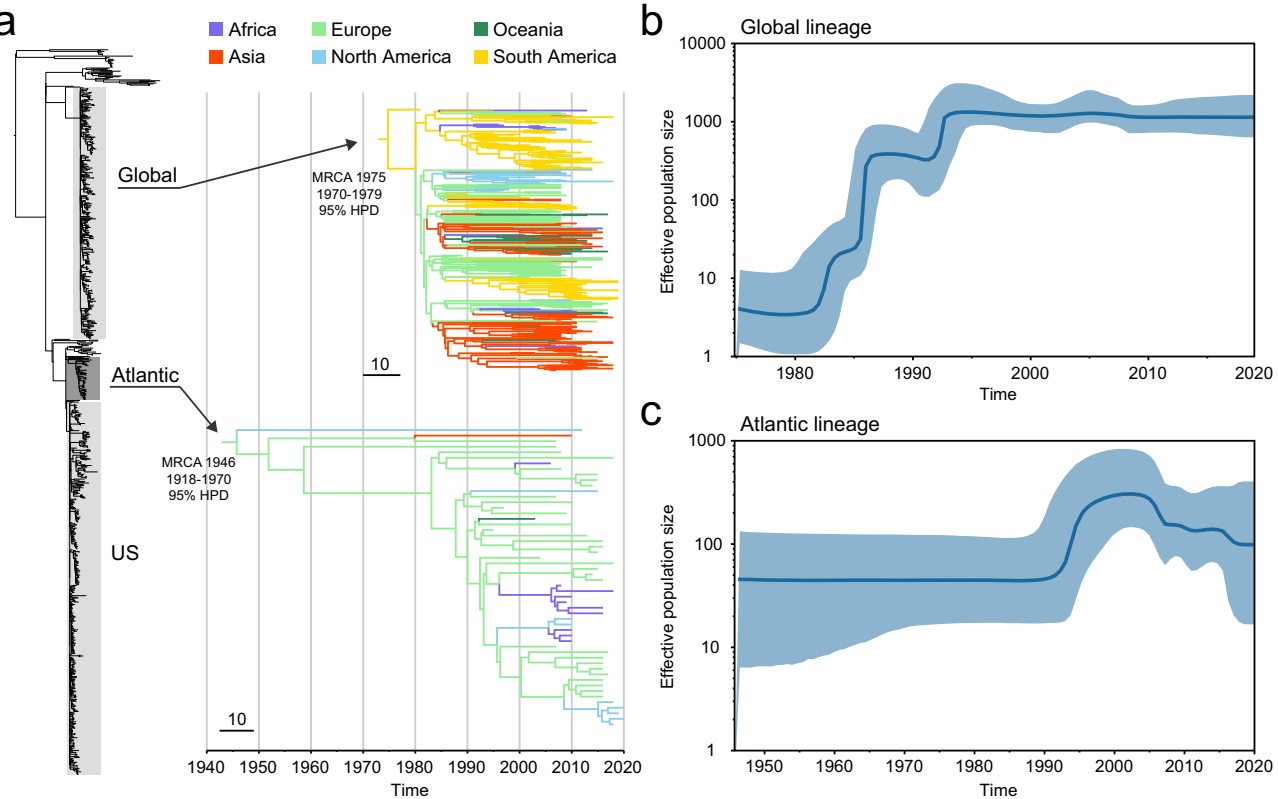

**Fig. 5 Phylodynamic analyses of the Global and the Atlantic lineages. a** Maximum clade credibility trees and phylogeographical inference. The inferred median MRCA age of each lineage is shown with 95% HPD. MRCA, most recent common ancestor; HPD, highest posterior density. Terminal branches are color coded by the isolates' continent of origin. Internal branches are colored to indicate the predicted geographical origin with the highest posterior probability value inferred using maximum-likelihood ancestral trait reconstruction. The scale bars indicate 10 SNPs. **b**, **c** Bayesian skyline plots showing the historical changes of the effective population size of poultry *Salmonella* Enteritidis populations in the Global and the Atlantic lineages. Blue lines represent the medians of estimated effective population sizes. Blue shadings indicate the upper and lower bounds of the 95% HPD intervals.

phylogeography of *Salmonella* Enteritidis dispersal, and temporal dynamics of *Salmonella* Enteritidis populations.

The most recent common ancestor (MRCA) of the Global lineage was dated to the late 1970s (Fig. 5a), closely preceding the beginning of significant merging and acquisition in the global breeding business in the early 1980s[8]. The MRCA of the Atlantic lineage was dated to the 1940s (Fig. 5a), coinciding with the beginning of modern poultry breeding on both sides of the Atlantic[5,33]. The US lineage likely emerged around a similar time because of its close phylogenetic relatedness to the Atlantic lineage (Fig. 4a). Bayesian phylogeographic inference based on ancestral state reconstruction suggests that the Atlantic lineage and the Global lineage represented by sampled isolates may be ancestrally traced to Europe and South America (Fig. 5a).

Model-based Bayesian estimation of effective population size showed population changes of poultry *Salmonella* Enteritidis in the Global and the Atlantic lineages. Rapid expansions of both lineages occurred in the 1980s (Fig. 5b) or the 1990s (Fig. 5c), coinciding with the *Salmonella* Enteritidis pandemic. While the Atlantic lineage started declining in the early-2000s (Fig. 5c), consistent with the observed decrease of *Salmonella* Enteritidis prevalence in the US and the UK since the late 1990s, the Global lineage has plateaued and remained steady since the mid-1990s (Fig. 5b).

**International trade of breeding stocks is concordant with global spread of *Salmonella* Enteritidis.** Historical data on international trade of live poultry are publicly available at Food and Agriculture Organization of the United Nation (FAO),

Observatory of Economic Complexity (OEC), and US Department of Agriculture Foreign Agricultural Service (FAS). FAO data are compiled in a single general category of live chickens traded from 1986 to 2019 (FAO data from 1961 to 1985 do not have importer information), the majority of which are conceivably breeding stocks. OEC data from 1995 to 2017 specify days-old birds under 185 grams of weight, most of which were presumably breeding stocks (OEC data from 1962 to 1994 are indistinguishable between chicken and non-chicken poultry). FAS data, which document US export, distinguish breeding stocks from other types of live chickens from 1989 to 2020. According to FAS data, US export of live chickens is overwhelmingly young birds under 185 grams (over 86.0%), in particular, broiler-type breeding stocks that accounted for 63.7% to 92.8% of the annual export (Supplementary Fig. 4). Similarly, comparison between OEC breeding stock data from 1995 to 2017 and FAO live chicken data in the same period showed that at least 94% of annual global trade of live chickens was breeding stocks.

From 1962 to 2019, a total of 219 countries were involved in the intercontinental trade of live poultry. It is evident that transcontinental movements of live poultry over the past six decades overwhelmingly originated from Europe and the US (Fig. 6a). In any given year, their combined export accounted for 84% to 98% of the intercontinental market size. From 1962 to 2019, the total intercontinental trade of live poultry grew over eight times after adjustment of inflation.

*Salmonella* Enteritidis isolates from 46 countries were included in our phylogenetic analysis, 23 of which were represented by poultry isolates (Fig. 4a). All these countries are importers of live

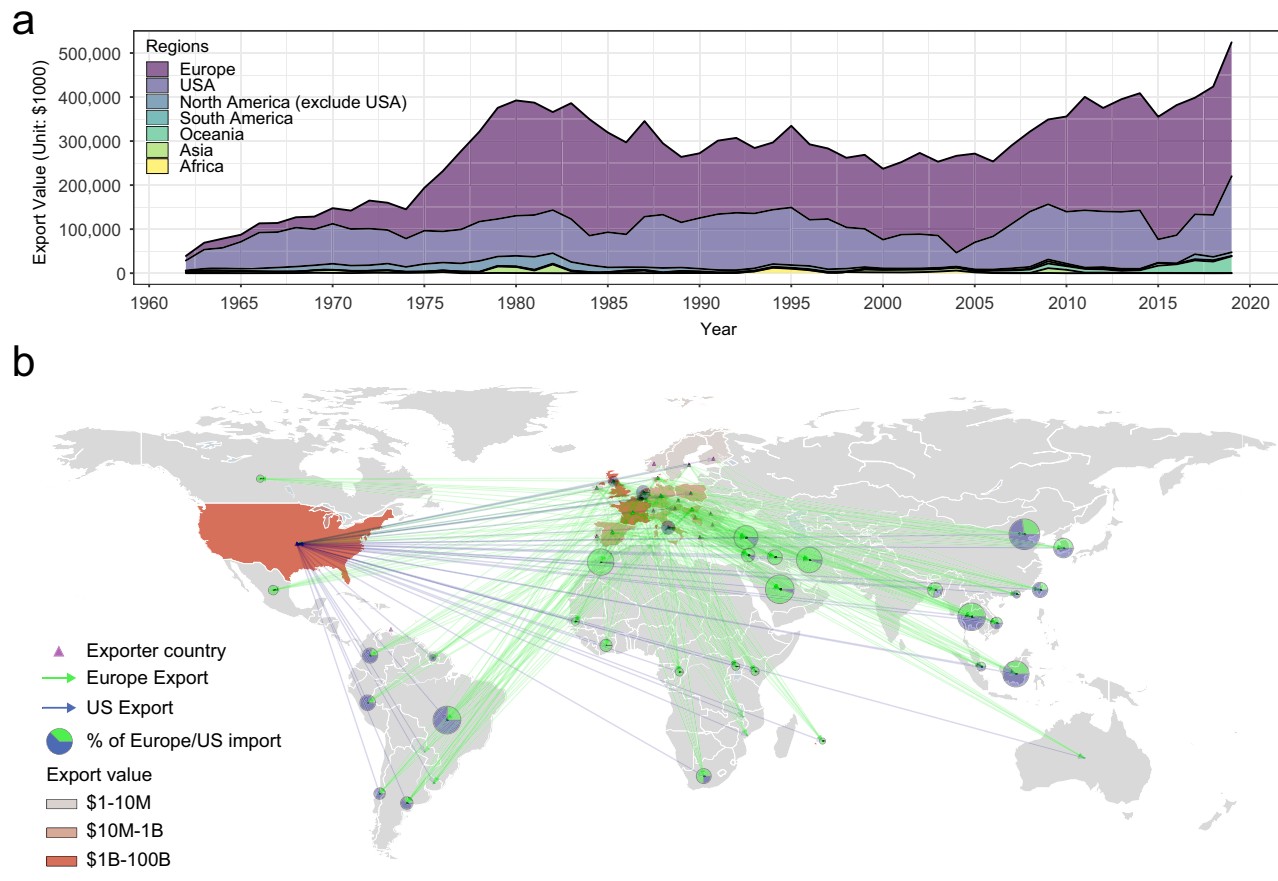

**Fig. 6 Intercontinental trade of live poultry. a** Share of intercontinental trade of live poultry by exporting continents or country from 1962 to 2019. Export between 1962 and 1985 is based on the trade data of live fowls at in Observatory of Economic Complexity database. Export between 1986 and 2019 is based on the trade of live chickens (*Gallus domesticus*) and guinea fowls (*Numida meleagris*) according to the Food and Agriculture Organization of the United Nations. **b** Cumulative intercontinental export of live poultry originated from Europe and the US between 1962 and 2019. Forty-six importers represented by isolates in the phylogenetic analysis (Fig. 4) are shown. Cumulative trade volumes between $24 million (Belgium) and $754 million (China mainland) are designated with pie charts showing the comparisons between total import values from Europe and from the US. The sizes of the pie charts are proportional to the volumes of trade.

poultry, or specifically breeding stocks (OEC data from 1995 to 2017), from Europe and/or the US (Fig. 6b).

**Quantitative assessment of breeding stocks as a driver of geographic dispersal of *Salmonella* Enteritidis.** We used the generalized linear model (GLM) extension of Bayesian phylogeographic inference[34] to quantify relative contributions of potential explanatory variables to the international spread of the Atlantic and the Global lineages; these lineages exhibited strong temporal signals (Figs. 4b and 5) optimal for phylogeographic modeling. The GLM approach tested these empirical factors associated with poultry trade by simultaneously reconstructing spatiotemporal history of the lineages using genetic data of *Salmonella* Enteritidis.

For the Global lineage, international trade of eggs including both hatching and unfertilized eggs yielded a high inclusion probability (0.89), considerably exceeding that of any other tested variables describing international poultry trade (Fig. 7). Inclusion probability reflects the posterior frequency at which the variable was included in the model and therefore represents quantitative support for the explanatory or predictive power of the variable[34]. The high inclusion probability, together with the positive mean and the narrow credible interval of the corresponding GLM

coefficients (Fig. 7), suggests a strong contribution of egg trade to the international dispersal of *Salmonella* Enteritidis.

While OEC trade data used for the analysis do not distinguish egg types, international trade of hatching eggs instead of unfertilized eggs was likely the major driver for the spread of the Global lineage to the sampled countries as suggested by the GLM analysis. First, the vast majority (84%) of poultry isolates included in the GLM analysis were isolated from chicken, which were potentially attributable to hatching eggs but unlikely from imported unfertilized eggs. Repeating the GLM analysis using only chicken isolates ($n = 107$) without egg isolates ($n = 23$) produced similar results (Fig. 7), except a higher contribution of exporter sample size due to unbalanced sampling of chicken isolates among exporter countries. This result further excluded unfertilized eggs as a causal factor in the observed dispersal of the Global lineage. Second, FAS data on US export specify egg types and show that significant proportions of internationally traded eggs, including those exported to the investigated countries, were hatching eggs used as breeding stocks (Supplementary Fig. 5). FAS data also show that hatching eggs accounted for substantial percentages of exported breeding stocks from the US to the investigated countries (Supplementary Fig. 5). Notably, two surveys in 1991 and 1998 in the US both found *Salmonella* on eggshells from primary breeder hatcheries[35].

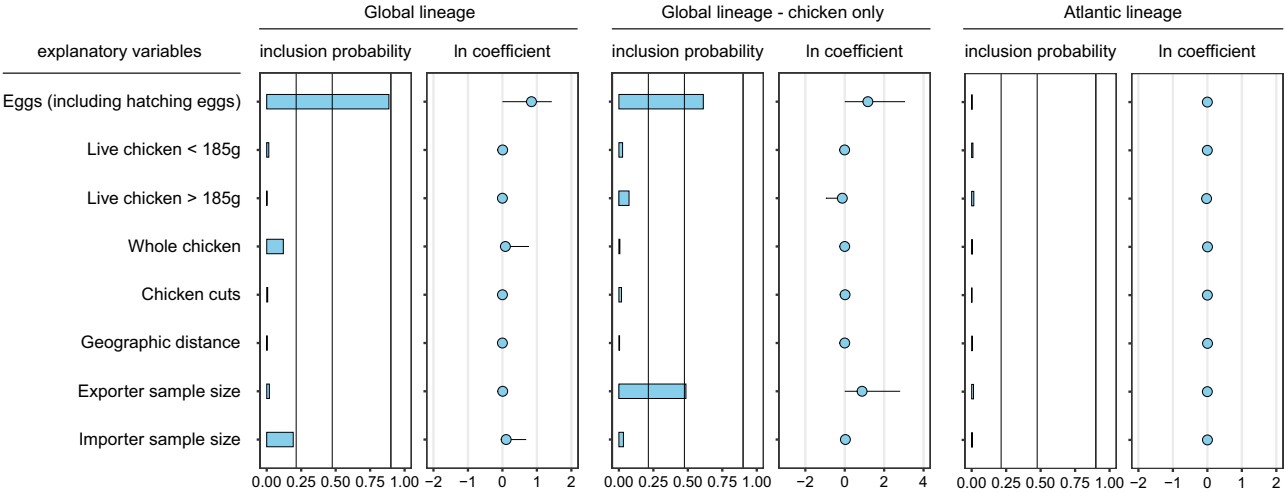

**Fig. 7 Contributions of explanatory variables to the international spread of *Salmonella* Enteritidis in the Global and the Atlantic lineages.** Inclusion probabilities corresponding to Bayes factor support values of 3, 10, and 100 are represented by vertical lines in the bar plots, respectively. The conditional effect size (ln coefficient), which is the effect size when the variable is included in the model, is represented by the mean and credible intervals (95% HPD intervals) of the GLM coefficients on a log scale. HPD, highest posterior density. Bars show the posterior probability of inclusion of each variable in the GLM model. Circles show the estimated conditional effect sizes for the GLM coefficients (>0 = positive association, <0 = negative association).

For the Atlantic lineage, none of the tested variables yielded a meaningful inclusion probability (Fig. 7), likely due to its small sample size being insufficient for robust GLM analysis. In the Atlantic lineage, only four countries were represented by at least five isolates, resulting a total of 29 isolates for analysis. In comparison, the Global lineage included 17 countries represented by 5 or more isolates, creating a total sample size of 275 isolates.

## Discussion

Traceback investigations combining trade data and pathogen genomes have resolved recent transmissions of *Salmonella* Enteritidis via eggs in Europe[36] and avian influenza via live poultry in China[37] at the scale of individual outbreaks. Here we present evidence that *Salmonella* Enteritidis dispersal on a pandemic scale may originate from the top of the poultry supply chain that has become globalized.

To maintain trade secrets and high levels of biosecurity of primary breeding, external inspection of breeding stocks is argued to be kept to the minimum[9]. Without *Salmonella* Enteritidis isolates directly from breeding stocks, it is difficult to conclusively trace *Salmonella* Enteritidis transmission to the top of the poultry supply chain. Nevertheless, the discovery of essentially identical *Salmonella* Enteritidis strains from broilers in Suriname and the US strongly supports the hypothesized intercontinental transmission of *Salmonella* Enteritidis via breeding stocks. First, the Suriname isolate was isolated from a domestically raised bird, precluding the possibility of its importation via processed poultry products. Second, the likely only overlap in poultry productions between the two countries is the sourcing of breeding stocks, with Suriname's breeding stocks predominantly supplied by the US. Finally, temporarily close isolations of these isolates (minimum of one year apart) further suggest their connection. Besides, the observation that other serotypes might have similarly spread to Suriname and other countries in the region provides additional support for the hypothesized mode of *Salmonella* Enteritidis dispersal.

The discovery also suggests that upon arrival on breeding stocks, the imported *Salmonella* Enteritidis could infiltrate and traverse the domestic poultry production system and eventually infect pre-harvest birds. Such a penetration may set the stage for sustained presence and proliferation of *Salmonella* Enteritidis in production flocks, if the imported strain persists in the production environment. Through the persistence, the exogenous strain may become endemic and cause rising human infections via contaminated poultry products as observed in many countries during the *Salmonella* Enteritidis pandemic. In Suriname, genomic surveillance of poultry *Salmonella* has not been established and only six *Salmonella* genomes of five serotypes mostly from 2016 were publicly available. The small sample size and limited timeframe made it difficult to determine the hypothesized domestic dissemination and persistence of imported *Salmonella*. Nevertheless, three of the six isolates can be linked to poultry isolates in the US (Supplementary Data 1), suggesting that routine surveillance may provide further evidence to test the hypothesis.

In the Global lineage, we found established and diverging clades in Chile and Brazil that were descended from closely related ancestral strains. These clades likely represent earlier instances of the postulated *Salmonella* Enteritidis introduction and establishment. The observation is consistent with the documented early emergence of *Salmonella* Enteritidis in both countries in the mid-1990s[38,39].

The recent sampling of the Suriname isolate (2016) raises a concern that breeding stock-mediated intercontinental spread of *Salmonella* Enteritidis still occurs. Similarly, a Mauritius clade in the Atlantic lineage supports a relatively recent *Salmonella* Enteritidis introduction to the country, which is in agreement with the pathogen's emergence and clonal dissemination in the country that was detected only after 2007[40].

Protracted spread of *Salmonella* Enteritidis via breeding stocks, which is often certified *Salmonella* Enteritidis-clean or *Salmonella* Enteritidis-monitored by the supplier in compliance with the US National Poultry Improvement Plan[41], may appear to contradict with the observed decline of *Salmonella* Enteritidis in the US and Europe since the late 1990s. However, the decline was attributed to interventions throughout poultry production[42], while the actual prevalence of *Salmonella* Enteritidis in breeding stocks remains little surveyed or published. Given the large volumes of breeding stock trade—the US alone exported ~56 million birds in 2019, a rare occurrence of *Salmonella* Enteritidis in breeding stocks, if introduced into poultry production, could be massively amplified (Fig. 1).

Beyond revealing the singular case of likely intercontinental dispersal of *Salmonella* Enteritidis, we further demonstrated that systemic spread of the pathogen likely originates from centralized origins. First, poultry *Salmonella* Enteritidis from a global sampling display limited genomic diversities compared with *Salmonella* Enteritidis from other sources, which can be explained by a few common origins of *Salmonella* Enteritidis circulating in poultry. In comparison, unrelated and dispersed origins of poultry *Salmonella* Enteritidis in different continents would result highly diverse *Salmonella* Enteritidis populations in poultry that reflect the complicated global evolutionary history of the serotype[43]. Second, phylodynamic analyses provide spatio-temporal support to the hypothesized central origins. The Global lineage appears to be a direct consequence and lingering concern of breeding stock-mediated *Salmonella* Enteritidis spread, as its estimated emergence closely preceded the start of global agglomeration of breeding stock supply and its estimated population size remains at a plateau level after the sharp increase during the pandemic. The Atlantic lineage was estimated to have emerged closer to the beginning of pedigree breeding. This lineage likely represents early poultry strains circulating in the US and Europe where industrialized poultry production first appeared. While its estimated population size declined since the early-2000s, reflecting the observed epidemiologic trend in the US and Europe, isolates in this lineage still persisted in poultry production and recently caused the largest salmonellosis outbreak in Europe[16]. Finally, qualitative support for the feasibility of widespread dispersal of *Salmonella* Enteritidis via breeding stocks was obtained as every country represented in the global phylogeny of poultry *Salmonella* Enteritidis imported breeding stocks from the US and/or Europe. Quantitative assessment that integrates phylodynamics of *Salmonella* Enteritidis and trade dynamics of breeding stocks further supports a driving role of breeding stock trade in shaping spatiotemporal dispersal of the Global lineage. Together, these findings corroborate the global scale of the hypothesized *Salmonella* Enteritidis spread.

Ancestral state reconstruction signaled a South American ancestor of the Global lineage, largely due to a few historical poultry isolates from Brazil that are basally positioned in the lineage. One may find the predicted origin at odds with the US and Europe being the earliest exporters of breeding stocks and the first places to report the rise of *Salmonella* Enteritidis. The seeming discrepancy may be reconciled by Brazil's early import of US breeds dating back to the 1940s and the 1950s[44]. The historical Brazilian isolates might have descended from US strains, which had arrived in Brazil before the pandemic. It should be noted that unbalanced datasets (e.g., an overrepresentation of European isolates in the Atlantic lineage) or a few historical isolates (e.g., early Brazilian isolates in the Global lineage) might skew ancestral state reconstruction. Despite the bias and the noise, both lineages were likely traceable to the US and/or Europe, which is corroborated by the fact that intercontinental trade of breeding stocks is mostly unidirectional from the US and Europe to other countries (Fig. 6).

The Global lineage and the combined Atlantic-US lineage defined in our study using poultry isolates are consistent with the "global epidemic clone" and the "global outlier cluster" described by Feasey et al.[45] using human isolates. Our phylogenetic analysis also shows common populations and similar strains of *Salmonella* Enteritidis between humans and chickens. Considering that poultry products are the primary vehicle of *Salmonella* Enteritidis transmission to humans, the *Salmonella* Enteritidis pandemic may ultimately be attributable to *Salmonella* Enteritidis-infected breeding stocks.

The historical puzzle surrounding the origin(s) of the pandemic is perplexed by the involvement of multiple strains of different phage types[17]. Such a polyclonal nature of the pandemic may confound the notion of centralized origins. Using WGS-based subtyping, we found that the Global and the Atlantic lineages each have isolates of different pre-determined phage types, some of which are phylogenetically close (Fig. 4a). WGS thus better resolved the relationship among circulating strains and showed their common evolutionary origins in few distinct lineages.

Another confounding factor to our findings is alternative routes for *Salmonella* to enter poultry production. Among the routes, feedstuff appears more plausible than others to spread *Salmonella* globally because of international trade of poultry feeds[46]. However, the speculated role of feed contamination in *Salmonella* Enteritidis spread was disputed by the lack of correlation between the serotypes found in poultry and those detected in feed[47]. Neither a US surveillance from 2002 to 2009[48] nor a Dutch survey from 1990 to 1991 found *Salmonella* Enteritidis in poultry feeds[49]. Specifically for the Suriname case, the country has not imported poultry feeds or feed components (except corn) from the US since 2012 (Supplementary Fig. 6). Migratory birds may also transmit pathogens over long distances. However, *Salmonella* Enteritidis is rarely found in wild birds[50]. Unlike *Salmonella* Enteritidis, *Salmonella* Typhimurium is established in both wild birds and poultry, but *Salmonella* Typhimurium lineages associated with the two sources were found distinct[51], suggesting unlikely transmission of *Salmonella* from wild birds to industrialized poultry production.

In conclusion, our findings shed light on a major foodborne risk at a critical, highly agglomerated, and less transparent section of the food chain. Despite decades of significant progress on *Salmonella* control in poultry, the evidence provided here calls for further investigation and potential intervention into the global spread of *Salmonella* from centralized origins at the pinnacle of poultry production.

## Methods

**Population structure and genomic diversity of *Salmonella* Enteritidis based on cgMLST**. We retrieved 33,142 *Salmonella* Enteritidis genomes that were available at EnteroBase[31] as of November 2020; serotype was confirmed by SeqSero2 v1.1.1[52]. Source of the isolates (e.g., poultry, human, soil/dust) was extracted from the "Source Type" column in EnteroBase. These *Salmonella* Enteritidis genomes belonged to 32 sources defined and documented in EnteroBase. Genomes without source information ($n = 2,930$) and cgST ($n = 131$) and genomes that had not been released ($n = 66$) were excluded from further analysis. This led to a final set of 30,015 genomes from 98 countries during 1949–2020. Minimum spanning trees were created based on cgSTs in EnteroBase using GrapeTree v1.5.1[53] with MSTree V2 algorithm. The cgMLST allele profiles were downloaded from EnterBase. Genomic diversity of the isolates within each source was assessed by calculating all pairwise allelic differences (PADs) between any two isolates. When multiple isolates shared the same cgST, one isolate was randomly picked and used for PAD calculation to ameliorate potential sampling biases caused by overrepresentation of closely related isolates.

**Phylogeny of poultry *Salmonella* Enteritidis isolates based on SNP**. We sampled a representative subset from the 30,015 genomes to build a phylogeny of poultry isolates under a broad phylogenetic context of *Salmonella* Enteritidis using SNP (Supplementary Data 2). For SNP analysis, raw sequencing reads of the sampled genomes were retrieved from Sequence Read Archive (SRA, https://www.ncbi.nlm.nih.gov/sra) or European Nucleotide Archive (ENA, https://www.ebi.ac.uk/ena/browser/home) using fastq-dump v2.10.8 of the SRA ToolKit[54].

To represent *Salmonella* Enteritidis from US poultry, we selected 608 genomes to include (1) historical poultry isolates that were collected no later than 2009 ($n = 210$), and (2) post-2009 isolates ($n = 398$) that evenly spanned a Neighbor-Joining tree of 3,528 genome assemblies of US poultry isolates in EnteroBase, covering all major clades of the tree. The Neighbor-Joining tree was built using Mashtree v1.1.2[55].

To represent the diversity of *Salmonella* Enteritidis in poultry outside the US and from other sources worldwide, we selected 1,040 genomes from poultry ($n = 603$), non-poultry avian ($n = 60$), wild and aquatic animals ($n = 187$), seafood ($n = 36$), water ($n = 148$), and feed ($n = 6$) in North America, Europe, South America, Asia, Africa, and Oceania. The Suriname isolate from a domestically raised bird was sampled by the Caribbean Integrated Surveillance System on Antimicrobial Resistance[56] and included here. We further sequenced *Salmonella* Enteritidis isolates from domestic poultry in Mauritius ($n = 8$)[40]. DNA was

extracted using the QiaAmp DNA Mini kit (Qiagen, Germantown, MD, USA). DNA concentrations were determined by the Qubit BR dsDNA assay kit (Invitrogen, Waltham, MA, USA). Libraries were prepared using the Illumina Nextera XT DNA Library preparation kit (Illumina, San Diego, CA, USA) and sequenced on a MiSeq sequencer (Illumina, San Diego, CA, USA) per manufacturer's instruction.

To represent epidemiologic, geographic and phylogenetic diversity of *Salmonella* Enteritidis circulating in humans, we selected *Salmonella* Enteritidis genomes from previous studies. These genomes included (1) recent outbreaks linked to eggs in Europe ($n = 52$)[16,57], (2) a US *Salmonella* Enteritidis survey ($n = 40$)[58], (3) a global *Salmonella* Enteritidis survey (global and African epidemic clades, $n = 48$)[45], and (4) genomes from Asia ($n = 33$), Oceania ($n = 24$) or South America ($n = 29$) in SRA as of December 2019. These continents were less represented by publicly available *Salmonella* Enteritidis genomes and the genomes were randomly selected from each continent.

From the selected genomes ($n = 1,882$), 154 were considered as low quality for having a genome assembly N50 size <100,000 or a sequencing coverage <30× according to pre-established criteria[51,59]. Another 10 were considered as outliers because they did not belong to the HC400_12 cluster defined by EnteroBase to which other selected isolates belonged and differed by at least 400 cgMLST alleles from other isolates in the cluster. The low quality and outlier genomes were excluded from phylogenetic analysis using SNP. Genome assemblies were generated by SPAdes v3.14.1[60] and evaluated with QUAST v4.5[61]. Prior to assembly, sequencing reads were trimmed and filtered using established methods[51]. Specifically, raw reads were processed by Trimmomatic v0.36[62]. The leading and the trailing three nucleotides were removed from the reads, and a four-nucleotide sliding window was applied to remove nucleotides from the 3' ends if the average Phred score within the sliding window <20. Reads <75 bp were discarded.

The remaining 1,718 genomes were used to build a SNP phylogeny. SNP calling was performed by SnapperDB v1.0.6[59] using default settings with P125109 genome (NCBI accession: AM933172) as the reference. Genome regions related to repetitive sequences and phages were removed from the reference genome before SNP detection and recombinant sequences were discarded from SNP alignment[51]. Repetitive and phage sequences were detected in the reference genome by MUMmer v4.0[63] and PHASTER[64], respectively. Gubbins v2.3.4[65] with default settings was used to detect recombinant sequences. The final alignment of core genome SNPs was used to build a maximum likelihood phylogenetic tree using PhyML version 20120412[66].

To alleviate sampling biases due to redundant genomes, we identified clusters of closely related isolates that had the same sample source, isolation year, and country of isolation and differed by <10 SNPs. One representative isolate was randomly selected from each cluster and kept. The rest of the cluster were considered redundant and discarded. A total of 804 redundant genomes were removed, leading to a final set of 914 genomes from 46 countries during 1954–2020 for phylodynamic analyses (Supplementary Data 2).

**Phylodynamic analyses.** Subtrees (i.e., internal nodes with >20 leaves) with strong temporal signals of SNP accumulation were identified throughout the *Salmonella* Enteritidis phylogeny using a custom Python 3.5 script[67]. Briefly, we screened all subtrees of the phylogeny by calculating the correlation coefficient between isolation years of the corresponding isolates and the tip-to-root distances of these isolates. Subtrees with squared coefficient ($R^2$) of either the Spearman's or the Pearson's correlation >0.4 were selected for further analysis. High-quality SNPs were called from genomes in each candidate subtree and maximum likelihood trees were built as described above. TempEst v1.5.3[68] was used to assess if there was sufficient temporal signal ($R^2 > 0.4$) for phylogenetic molecular clock analysis. After removing five outliers that substantially deviated from other genomes showing temporal signals, 317 genomes in the Global lineage and 54 genomes in the Atlantic lineages were subjected to model-based MRCA dating, phylogeographical reconstruction, and population dynamics analysis using BEAST v1.10.4[69]. Specifically, multiple BEAST analyses were performed on alignments of concatenated SNP of the candidate subtrees using constant population size, exponential growth, and Bayesian skyline models, in combination with a strict molecular clock or a relaxed log-normal molecular clock, to identify the model that best fitted the data. For each of the model combinations, three independent chains of 100 million generations with sampling at every 10,000 iterations were performed to ensure convergence. The three runs were then combined by LogCombiner v1.10.4[70] with the first 10 million steps from each run treated as burn-in. For each of the runs, the marginal likelihood was calculated using path sampling and stepping-stone sampling[71] and was used to assess model performance. In all cases, the combination of Bayesian skyline model and relaxed log-normal molecular clock was favored by yielding the highest Bayes factor (BF) among model combinations (Supplementary Data 3). Maximum clade credibility trees were generated by TreeAnnotator v1.10.4 within the BEAST platform[69]. The ages of MRCA and substitution rates were estimated by Tracer v1.7.1 within the BEAST platform[69] (Supplementary Data 4). Estimates were reported as median values with 95% highest posterior density (HPDs). Posterior probability values were used as support for ancestral nodes and their estimated geographical locations. The Bayesian skyline plot was made using Tracer v1.7.1 to estimate changes in the effective population size over time.

**Trade data.** Data for international trade of live poultry were obtained from three sources: Food and Agriculture Organization (FAO), Observatory of Economic Complexity (OEC), and USDA Foreign Agricultural Service (FAS) (Supplementary Data 5). Historical trade values were adjusted for inflation and expressed in constant 2019 U.S. dollars by US Inflation Calculator[72], which is based on the US government Consumer Price Index (CPI) data published in 2021. Custom scripts[73] using ggplot2 v3.3.3 in R v3.5.1 and Matplotlib Basemap Toolkit v1.2.0 in Python 3.5 were used to visualize intercontinental trade of breeding stocks and live poultry.

**General linear model (GLM) extension of Bayesian phylogeographic inference.** Potential explanatory variables for *Salmonella* Enteritidis dispersal were quantitatively assessed using the generalized linear model extension of Bayesian phylogeographic inference as previously described[34]. Variables tested include the trade of eggs (both hatching and unfertilized eggs), live chicken over 185 g, live chickens under 185 g (day old birds used as breeding stocks), whole chickens (processed), chicken cuts, geographic distance between trading countries (defined as the shortest distance between the capital cities), importer sample size, and exporter sample size. All the trade data were obtained from OEC. Between any two investigated countries, the aggregated value of the traded commodity from 1995 to 2017 was used for the GLM analysis. The analysis was separately performed on the Atlantic lineage that includes 29 *Salmonella* Enteritidis isolates from the US, the UK, Poland, and Italy; and on the Global lineage that includes 275 *Salmonella* Enteritidis isolates from Brazil, the UK, Chile, South Korea, China, Australia, the USA, Thailand, Colombia, Argentina, Austria, Canada, Malawi, Poland, Lebanon, South Africa, and Singapore. Only countries represented by a minimum of five isolates were selected for analysis[37]. To preserve the temporal signals necessary for phylogeographic analysis, non-poultry isolates were also included, but >50% of isolates used for modeling in both lineages were isolated from poultry. BEAST analyses were performed using the Bayesian skyline model in combination with a relaxed log-normal molecular clock as described above.

**Reporting summary.** Further information on research design is available in the Nature Research Reporting Summary linked to this article.

## Data availability

All data generated or analyzed during this study are provided in this published article and its supplementary information sites. The entire set of 33,142 available *Salmonella* Enteritidis genomes at EnteroBase as of November 2020 is available at https://enterobase.warwick.ac.uk/species/senterica/search_strains?query=workspace:49557. The *Salmonella* Enteritidis genomes from Mauritius sequenced in this study have been deposited in SRA database under accession codes SRR13681353, SRR13681354, SRR13681355, SRR13681356, SRR13681357, SRR13681358, SRR13681359, and SRR13681360. Accession numbers and metadata of genomes that were used for building *Salmonella* Enteritidis phylogeny using SNP are available at Supplementary Data 2. *Salmonella* cgMLST allelic profile is available at http://enterobase.warwick.ac.uk/schemes/Salmonella.cgMLSTv2/profiles.list.gz. An interactive minimum spanning tree of 30,015 *Salmonella* Enteritidis isolates (Fig. 3a) is available at https://enterobase.warwick.ac.uk/ms_tree/51520. An interactive minimum spanning tree of 3,449 *Salmonella* Enteritidis poultry isolates (Fig. 3c) is available at https://enterobase.warwick.ac.uk/ms_tree/50727. International trade data of live poultry and poultry products are available from three sources: Food and Agriculture Organization (FAO, http://www.fao.org/faostat/en/#data/TM), Observatory of Economic Complexity (OEC, https://legacy.oec.world/en/resources/data/), and USDA Foreign Agricultural Service (FAS, https://dataweb.usitc.gov/). Trade data of live poultry from FAO were obtained by searching for entries with SITC number 1057. Trade data of live poultry from OEC were obtained by searching for entries with HT92 numbers 010511 and 010591. Trade data of bird eggs from OEC were obtained by searching for entries with HT92 number 040700. Trade data of live poultry from FAS were obtained by searching for entries with HTS numbers 0105.11.0010, 0105.11.0020, 0105.11.0040, and 0105.91.0000. Trade data of fertilized eggs from FAS were obtained by searching for entries with HTS number 0407.11.0000.

## Code availability

Custom code for screening temporal signals throughout a phylogeny (Fig. 4b)[67] is available at https://doi.org/10.5281/zenodo.5142197. Custom codes for visualization of global live poultry trade (Fig. 6)[73] are available at https://doi.org/10.5281/zenodo.5142417.

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

## Acknowledgements

Most of publicly available *Salmonella* genomes used in the study were sequenced and/or deposited by GenomeTrakr, PulseNet, the NARMS program, USDA Food Safety and Inspection Service, Public Health England, and EnteroBase. We thank the support of Caribbean Integrated Surveillance System on Antimicrobial Resistance in Agriculture (CISARA) project by the 10th European Development Fund SPS Project, the Inter-American Institute for Cooperation on Agriculture (IICA), and the Suriname Government. We thank Mark Achtman and Zhemin Zhou for valuable help on EnteroBase. We thank Patti Fields and Francisco Diez-Gonzales for reviewing the manuscript. This study was supported in part by a Hatch project (1006141) from the USDA National Institute of Food and Agriculture.

## Author contributions

L.S. performed bioinformatics, genomics, and phylodynamics analyses. Y.H. reviewed literature on *Salmonella* in poultry feed and performed analyses on global trade of live poultry and feed. D.A.M. performed whole-genome sequencing. X.D. conceptualized the study, designed analyses, interpreted results, and wrote the original draft. All authors reviewed the paper.

## Competing interests

The authors declare no competing interests.
