## [Peer Review File · Nature Communications]

REVIEWER COMMENTS

Reviewer #1 (Remarks to the Author):

Overall:

The authors present an interesting paper which is the result of a large amount of work. The topic of global dissemination of Salmonella is relevant and timely.

The methods used are sound.

he discussion is on the long side.

Major comment:

I find the manuscript a bit on the speculative side, with large part of the evidence provided hanging on one isolate from Suriname. I am not saying that the poultry trade might not be to blame for the spread of SE, but the evidence is not very compelling, it feels still as conjecture evidence.

Minor comments:

Line 128-130: What is meant by isolate here? Did you use the wgMLST? Or do you mean that you have used the full dataset, as opposed to the second approach where you used only distinct cg types? This section might benefit from a clearer explanation.

Line 137-141: the parallel distributions in PAD in my opinion does not say anything per se on the relatedness of both sub-populations.

Line 167-168: I understand what you mean, but the way you explain it sounds as the isolation year is an attribute of the internal nodes, which is, of course, not the case. Please reformulate.

Line 179: Yes, because you have an overwhelming majority of European isolates in clade Atlantic. You might get different results if you would have a balanced subset of all locations.

And, how does this fit with your hypothesis of the Suriname isolate originating from US, or with the trade that seems to be unidirectional US-South America, or Europe-South America?

Line 230-231: sentence really unclear to me.

Line 241-243: They still remain isolated cases, which in colloquial speaking will increase that "likelihood". However, you do not integrate that into a real likelihood estimate.

Line 244-245: I guess if it were so, you might expect to find a bigger overlap between the two populations. I have to admit, however, that the sampling effort might not be comparable in between the various regions. Might still be useful to discuss.

Line 271: I would say they provide support for bacterial population size changes. One cannot exclude the possibility of increased pop size due to congruent changes in the agricultural practices around the world.

Line 272: Why is that? You have just shown that it emerged from South America, while you hypothesize that the stock-mediated spread occurred primarily from US and EU.

Line 287: Yes, but then how much should one rely on the ancestral state reconstruction?

Line 296-298: I am not sure I can see the phage types in the indicated figure, nor elsewhere in the results.

Line 311-321: not sure how this fits in the discussion / is off added value.

Line 379: What did you concatenate precisely?

Reviewer #2 (Remarks to the Author):

The authors present an interesting genomic analysis of Salmonella Enteritidis (SE) with a focus on understanding the pandemic that occurred in the 1980s. They attempt to reconcile the phylogenetic analysis with information on poultry trade to make assertions about the dispersal of the SE pandemic across the globe.

The central hypothesis is that centralized sourcing and international trade of SE-infected breeding stock is the most parsimonious explanation for SEs global emergence.

The integration of genomic data and trade data is an innovative approach and an important one to understand emergence of zoonotic pathogens.

The WGS analysis is well thought out and comprehensive and the description of the SE population on this scale is certainly an advance in the field.

Major Comments:

A significant drawback to this paper is that no formal attempt was made to compare the trade data to the phylogenetic data. It would have been a significant improvement if this data could have been compared in a statistical model. i.e. correlation between ancestral state reconstruction and poultry import / export dynamics.

Other comments

1) The authors identified several highly related poultry matches from different countries suggesting trade of contaminated breeding stock. They suggest that KDE of PADs suggest overlapping populations between human and poultry. This assertion in fact suggests that there is a similar variability / clustering in the human and poultry genomes not that they overlap. It would be of use to add a human vs poultry PAD plot to see if that gives the same distribution. This could also be tested with k-test.

2) Phylodynamic reconstruction of a global SE population was also performed. They identify three major poultry lineages, Global, Atlantic and US. Mixture between broiler and layers needs some more thought on this global time-scale. A section describing this mixing in terms of PADs would be welcome and perhaps could warrant an ancestral state reconstruction (broiler Vs layer) of its own. Phylodynamic analysis was used to date the Global and Atlantic lineages and ancestral state reconstruction performed on location.

REVIEWER COMMENTS

Reviewer #1 (Remarks to the Author):

Overall:

The authors present an interesting paper which is the result of a large amount of work. The topic of global dissemination of Salmonella is relevant and timely.

The methods used are sound.

The discussion is on the long side.

Major comment:

I find the manuscript a bit on the speculative side, with large part of the evidence provided hanging on one isolate from Suriname. I am not saying that the poultry trade might not be to blame for the spread of SE, but the evidence is not very compelling, it feels still as conjecture evidence.

We appreciate this remark and understand the reviewer's perception. When we found the Suriname isolate, we had a similar question: Was it a sporadic case or an embodiment of global dispersal? The bulk of the study was our attempt to make scientifically sound inferences and generalizations from the Suriname isolate, other similar sporadic isolates, global populations of SE, and international trade of live chicken and eggs.

Using the Suriname SE isolate, we attempted to establish the feasibility of the hypothesized intercontinental spread of SE. We believe the Suriname isolate adequately served the purpose.

In addition to this isolate, we would like to point out one serotype Ohio isolate from Suriname and two serotype Kentucky isolates from Barbados and Trinidad and Tobago as we described in the section "**Isolates of additional serotypes supporting intercontinental transmission via poultry**". These additional isolates also provide support for the feasibility. In particular, just like the Suriname isolate, these isolates were genetically highly close to recent US isolates, from domestically raised birds in these countries, isolated recently and within 1-3 years apart from their US counterparts, and originated from countries that predominantly imported breeding stock from the US (Supplementary Fig 2).

Additional but less conclusive evidence came from the isolates summarized in Table S2. As part of the revision, we performed additional analysis on the breeding stock trade between the linked countries (Supplementary Fig 1). We added this analysis to the manuscript along with the caution that an alternative vehicle of dispersal (import of processed poultry products) is possible:

"Trade of breeding stock was identified between the linked countries prior to the isolation of the isolates (Supplementary Fig. 1), suggesting the possibility of breeding stock as a dispersal vehicle of SE. However, processed poultry products were also traded between the linked countries during the same time span (Supplementary Fig. 1)"

After establishing the feasibility using sporadic isolates, we sought to reconstruct the timeline and infer the scope of the hypothesized dispersal to evaluate if it was spatially sufficient and temporally possible to cause the SE pandemic. Similar to other studies that probed the origins of historical pandemics (e.g., 1918 influenza virus [1] and plague [2]), such inferences rely heavily on population and evolutionary analyses, as we described in the sections of **“Close relatedness of SE from global poultry”** and **“Phylodynamics of circulating SE lineages.”**

Finally, in the section of **“Global trade of breeding stock”**, we showed that the global trade of the commodity in the past 50 years was concordant with our hypothesis. As suggested by Reviewer 2, we further integrated phylodynamics of SE and trade dynamics of breeding stock by using a phylogeographic inference model that had been recently applied to test potential drivers of viral pathogen dispersal [3, 4]. By quantitatively evaluating a range of potential explanatory variables, we provided statistical support that the trade of breeding stock (hatching eggs) likely drove the international spread of the Global lineage. Please see details in the new Results section **“Quantitative assessment of breeding stock as a driver of geographic dispersal of SE”**.

We believe our evidence is multifaceted and adequate to support our central hypothesis that *“centralized sourcing and international trade of SE-infected breeding stock is a parsimonious explanation for the synchronized and expansive spread of SE.”*

We acknowledged that *“without SE isolates directly from breeding stock, it is difficult to conclusively trace SE transmission to the top of the poultry supply chain.”* More compelling and direct evidence would require matched isolates from breeding stock at its origin and from domestic poultry of an importer country. We explained in Discussion that the lack of public scrutiny of primary breeding and inaccessible trade data from specific primary breeders makes such “smoking gun” evidence likely beyond reach.

Therefore, we believe it is justified to publish our hypothesis, which has long been speculated in the field (Line 61-71), is further supported by multifaceted and scientifically sound evidence from our study, and will help draw attention to a long lasting but still timely public health issue. As the title of our manuscript suggests, we intend to present evidence instead of jumping to a conclusion. As we concluded at the end of the paper, we hope the evidence will lead to further investigation and potential intervention into this issue.

We also would like explain that we applied highly stringent criteria when looking for sporadic SE isolates as evidence to implicate breeding stock for SE dispersal. Our criteria include:

- 1) The isolate has to be sampled from domestically raised bird instead of processed poultry products, because international trade of poultry products is common and may also disperse SE globally.
- 2) The importer country has to import breeding stock solely or overwhelmingly from a single exporter country shortly (5 years) before the isolation of the isolate to avoid the confounding factor that the isolate may come from a third country.

The first criterion excludes many isolates whose source information is just poultry in the public domain (e.g. isolates in Supplementary Table 2). We confirmed the domestic broiler origin of the Suriname isolate by tracing it back to the Ministry of Agriculture, Animal Husbandry and Fisheries of Suriname. We were fortunate to come upon this isolate because of a recent Salmonella survey in live poultry in the Caribbean region for an unrelated purpose of AMR monitoring (see Acknowledgment). Regrettably, our efforts to pinpoint the origin of poultry isolates from other countries were not as successful. In most cases, the submitter of the Salmonella genomes is a government agency that was restricted from sharing more information.

The second criterion narrows our attention to countries with an emerging domestic poultry sector whose history and supply chain of sourcing live poultry is not too complex to pinpoint the origin of imported breeding stock. One may expect that countries such as Canada and EU members would provide more isolates as evidence for SE dispersal via breeding stock. However, these countries have been importing breeding stock from multiple exporters over decades, making conclusive traceback difficult as we showed in a new supplementary figure (Supplementary Fig. 1).

Minor comments:

Line 128-130: What is meant by isolate here? Did you use the wgMLST? Or do you mean that you have used the full dataset, as opposed to the second approach where you used only distinct cg types? This section might benefit from a clearer explanation.

We greatly appreciate this comment as well as the reviewer's later comment on the discussion about the two approaches for PAD calculation. Both comments prompted us to re-examine this analysis. We eventually found that our claim of the sampling bias mitigating effect by one of the approaches was erroneous -- it was due to an artefact caused by downloading cgSTS of human isolates from Enterobase in different batches.

Specifically, because of the large number of human isolates at Enterobase, we had to download their cgSTS in two batches. We failed to detect that Enterobase did not keep the order of some cgMLST loci consistent across different batches of downloaded cgSTS. This inconsistency in loci order created two artificial "populations" of SE (i.e., two separately downloaded batches) featuring large pairwise allelic differences. After removing this artefact, the intra-source median PADs by using all isolates and by using unique cgSTS (i.e., down-sampling isolates of identical cgSTS) were similar for human isolates.

Therefore, we revised this part and deleted corresponding statements in Discussion, which the reviewer commented as "not sure how this fits in the discussion / is off added value". The artefact caused by the batch effect neither affected isolates from other sources (they were downloaded in single batches) nor changed our conclusion on median PAD of poultry isolates being the lowest among major SE sources.

Line 137-141: the parallel distributions in PAD in my opinion does not say anything per se on the relatedness of both sub-populations.

We deleted the statement *“This observation signals distinct SE populations that commonly circulate in both humans and chickens”*. While related populations can result parallel distributions in PAD, making inference the other way around is not appropriate without a formal proof.

Line 167-168: I understand what you mean, but the way you explain it sounds as the isolation year is an attribute of the internal nodes, which is, of course, not the case. Please reformulate.

As suggested, we reformulated the sentence:

“We performed an exhaustive search for temporal signals of SNP accumulation by screening every internal node of the phylogeny (i.e., inferred common ancestors) for strong correlation between isolation years of and branch lengths to the corresponding tips (i.e., extant isolates)” of the phylogeny.”

Line 179: Yes, because you have an overwhelming majority of European isolates in clade Atlantic. You might get different results if you would have a balanced subset of all locations. And, how does this fit with your hypothesis of the Suriname isolate originating from US, or with the trade that seems to be unidirectional US-South America, or Europe-South America?

We agree with the reviewer. We revised the statement:

“...suggests that the Atlantic lineage and the Global lineage represented by sampled isolates may be ancestrally traced to Europe and South America”

Our hypothesis of the Suriname isolate originating from US is supported by extant isolates sampled from two countries from 2016 to 2020 being genetically almost identical. This hypothesis is also supported by Suriname overwhelmingly relied on US export for breeding stock supply prior to and during this period.

We agree with the reviewer that a balanced dataset that better represents each country is ideal for ancestral state reconstruction. Using an unbalanced datasets limited by available isolates, we cautiously interpreted inferred ancestral states along with corroboratory evidence, especially as the reviewer suggested that breeding stock trade is mostly unidirectional from US and Europe to other continents (Fig 6).

We added to Discussion the statement below to address the potential pitfall of ancestral state reconstruction:

“It should be noted unbalanced datasets (e.g., an overrepresentation of European isolates in the Atlantic lineage) or a few historical isolates (e.g., early Brazilian isolates in the Global lineage) might skew ancestral state reconstruction. Despite the bias and the noise, both lineages were likely traceable to the US and/or Europe, which is corroborated by the fact that intercontinental trade of breeding stock is mostly unidirectional from the US and Europe to other countries (Fig. 6)”

We also would like to clarify that the Suriname isolate belongs to the US lineage:

“...and an overwhelmingly US lineage (US), which has the aforementioned Suriname isolate (Fig. 4a).”

Evidence to support our hypothesis that the Suriname isolate originating from the US does not come from ancestral state inference, which was performed on the Atlantic and the Global lineages but not on the US lineage (the US lineage did not exhibit strong temporal signal to support the inference and its ancestral state is most likely the US). The genomic evidence for the origin of the Suriname isolate is more direct and compelling as it is genetically almost identical to multiple US isolates.

Line 230-231: sentence really unclear to me.

We revised the sentence:

"Here we present evidence that Salmonella dispersal at the pandemic scale may originate from the top of the poultry supply chain that has global implication."

Line 241-243: They still remain isolated cases, which in colloquial speaking will increase that "likelihood". However, you do not integrate that into a real likelihood estimate.

We revised the sentence:

"Besides, the observation that other serotypes might have similarly spread to Suriname and other countries in the region provides additional support for the hypothesized mode of SE dispersal."

Line 244-245: I guess if it were so, you might expect to find a bigger overlap between the two populations. I have to admit, however, that the sampling effort might not be comparable in between the various regions. Might still be useful to discuss.

Thanks for the suggestion. We added the sentences below as suggested. Indeed, a substantial overlap should be expected, but that is only observable if sufficient SE samples are available in both exporter and importer countries.

"In Suriname, genomic surveillance of poultry Salmonella has not been established and only 6 Salmonella genomes of 5 serotypes mostly from 2016 were publicly available. The small sample size and limited timeframe made it difficult to determine the hypothesized domestic dissemination and persistence of imported Salmonella. Nevertheless, 3 of the 6 isolates can be linked to poultry isolates in the US (Supplementary Table 1), suggesting that routine surveillance may provide further evidence to test the hypothesis."

Line 271: I would say they provide support for bacterial population size changes. One cannot exclude the possibility of increased pop size due to congruent changes in the agricultural practices around the world.

We would like to clarify that bacterial population size changes was directly inferred by phylodynamic analyses.

We completely agree on the possibility of congruent changes in agricultural practices around the world resulting in increased population sizes of SE. We believe that centralized sourcing and international trade of breeding stock is part of the agricultural practices and played an important role in causing increased population sizes of SE inferred by phylodynamic analyses.

As we mentioned in the manuscript:

“The emergence of breeding selection and specialized breeders heralded structural transformation and consolidation of the poultry industry, which evolved into one of the most integrated agribusinesses as the major poultry markets in the US and Europe matured in the 1980s and the early 1990s”

“The Global lineage appears to be a direct consequence and lasting concern of breeding stock-mediated SE spread, as its estimated emergence closely preceded the start of global agglomeration of breeding stock supply and its estimated population size remains at a plateau level after the sharp increase during the pandemic.”

Line 272: Why is that? You have just shown that it emerged from South America, while you hypothesize that the stock-mediated spread occurred primarily from US and EU.

We explained in the next paragraph:

“Ancestral state reconstruction signaled a South American ancestor of the Global lineage, largely due to a few historical poultry isolates from Brazil that are basally positioned in the lineage. One may find the predicted origin at odds with the US and Europe being the earliest exporters of breeding stock and the first places to report the rise of SE. The seeming discrepancy may be reconciled by Brazil’s early import of US breeds dating back to the 1940s and the 1950s. The historical Brazilian isolates might have descended from US strains, which had arrived in Brazil before the pandemic.”

Please also see our reply to the next comment below.

Line 287: Yes, but then how much should one rely on the ancestral state reconstruction?

As we replied to an earlier comment (Line 179), we expanded the discussion here to address the potential pitfall of unbalanced data:

“It should be noted unbalanced datasets (e.g., an overrepresentation of European isolates in the Atlantic lineage) or a few historical isolates (e.g., early Brazilian isolates in the Global lineage) might skew ancestral state reconstruction. Despite the bias and the noise, both lineages were likely traceable to the US and/or Europe, which is corroborated by the fact that intercontinental trade of breeding stock is mostly unidirectional from the US and Europe to other countries (Fig. 6)”

Line 296-298: I am not sure I can see the phage types in the indicated figure, nor elsewhere in the results.

We added to the Results:

“For 8 isolates representing all 3 major lineages, their phage types had been pre-determined and published. These isolates and their phage types were denoted in Fig 4A and Table S1.”

Line 311-321: not sure how this fits in the discussion / is off added value.

Please see our reply to an earlier comment (Line 128-130). Again, we are very grateful for this comment because the reviewer's doubt led us to identify an erroneous claim due to a previously undetected artefact.

Line 379: What did you concatenate precisely?

We changed it to "*alignments of concatenated SNPs*".

References

1. Taubenberger, J., Reid, A., Lourens, R. *et al.* Characterization of the 1918 influenza virus polymerase genes. *Nature* 437, 889–893 (2005). <https://doi.org/10.1038/nature04230>
2. Morelli, G., Song, Y., Mazzoni, C. *et al.* *Yersinia pestis* genome sequencing identifies patterns of global phylogenetic diversity. *Nat Genet* 42, 1140–1143 (2010). <https://doi.org/10.1038/ng.705>
3. Lemey, P., Rambaut, A., Bedford, T. *et al.* Unifying Viral Genetics and Human Transportation Data to Predict the Global Transmission Dynamics of Human Influenza H3N2. *PLoS Pathogens* (2014). <https://doi.org/10.1371/journal.ppat.1003932>
4. Yang, Q., Zhao, X., Lemey, P. *et al.* Assessing the role of live poultry trade in community-structured transmission of avian influenza in China. *PNAS* 117 (2020). <https://doi.org/10.1073/pnas.1906954117>

Reviewer #2 (Remarks to the Author):

The authors present an interesting genomic analysis of *Salmonella* Enteritidis (SE) with a focus on understanding the pandemic that occurred in the 1980s. They attempt to reconcile the phylogenetic analysis with information on poultry trade to make assertions about the dispersal of the SE pandemic across the globe.

The central hypothesis is that centralized sourcing and international trade of SE-infected breeding stock is the most parsimonious explanation for SEs global emergence.

The integration of genomic data and trade data is an innovative approach and an important one to understand emergence of zoonotic pathogens.

The WGS analysis is well thought out and comprehensive and the description of the SE population on this scale is certainly an advance in the field.

Major Comments:

A significant drawback to this paper is that no formal attempt was made to compare the trade data to the phylogenetic data. It would have been a significant improvement if this data could

have been compared in a statistical model. i.e. correlation between ancestral state reconstruction and poultry import / export dynamics.

We greatly appreciate this comment. As suggested by the reviewer, we used the GLM approach developed by Lemey et al. [1] to quantitatively test the contribution of potential predictors of the spatial spread of SE. This model allows the integration of empirical data (e.g., breeding stock trade) with phylogenetic data (e.g., ancestral state reconstruction) as the reviewer suggested. Please see details of this analysis in the new Results section “**Quantitative assessment of breeding stock as a driver of geographic dispersal of SE**” as well as corresponding parts added to Discussion and Methods.

This formal analysis provides strong quantitative support that the trade of breeding stock in the form of hatching eggs served as a major driver for the dispersal of the Global lineage. We are very grateful for the reviewer’s suggestion because it allowed us to further integrate phylodynamics of SE with trade dynamics of breeding stock, which led to stronger evidence to support our central hypothesis.

Other comments

1) The authors identified several highly related poultry matches from different countries suggesting trade of contaminated breeding stock. They suggest that KDE of PADs suggest overlapping populations between human and poultry. This assertion in fact suggests that there is a similar variability / clustering in the human and poultry genomes not that they overlap. It would be of use to add a human vs poultry PAD plot to see if that gives the same distribution. This could also be tested with k-test.

We agreed with the reviewer and deleted the statement “*This observation signals distinct SE populations that commonly circulate in both humans and chickens*”.

First, it is more straightforward to show related SE clades circulating in humans and poultry from the phylogeny (Fig 4a), as we stated later in the manuscript:

“Human isolates included in this analysis intermingle with poultry isolates in all three lineages,”

“The Global lineage and the combined Atlantic-US lineage defined in our study using poultry isolates are consistent with the “global epidemic clone” and the “global outlier cluster” described by Feasey et al. using human isolates⁴⁴”.

Second, as noted by Reviewer 1, we realized that while strongly related populations can result parallel distributions of PADs, making inference the other way around is not appropriate without a formal proof.

As suggested, we did a human vs poultry PAD plot (Human-vs-Poultry includes PADs of all human-poultry isolates pairs) and arrived at a parallel distribution:

We decided not to add it to the manuscript, because it may be viewed as an over-interpretation of the observed parallel distribution of PADs, as Reviewer 1 commented and we explained above.

2) Phylodynamic reconstruction of a global SE population was also performed. They identify three major poultry lineages, Global, Atlantic and US. Mixture between broiler and layers needs some more thought on this global time-scale. A section describing this mixing in terms of PADs would be welcome and perhaps could warrant an ancestral state reconstruction (broiler Vs layer) of its own. Phylodynamic analysis was used to date the Global and Atlantic lineages and ancestral state reconstruction performed on location.

As suggested, we added the following section about chicken and egg isolates:

“Within poultry, the mixture of isolates from chicken and eggs suggests indistinguishable SE populations circulating in both broilers and layers (Fig. 4a). For each egg isolate on the phylogeny, the closest chicken isolate was 0 to 125 allelic differences (median: 17) or 0 to 101 SNPs (median: 15) away. For chicken and egg isolates in the Global lineage that exhibited strong temporal signals for robust ancestral state inference, it was common to observe egg isolates with an inferred chicken ancestor and vice versa, oftentimes with such ancestors and their progenies separating by less than 5 years (Supplementary Fig. 3).”

We also examined PAD distributions of eggs and chicken isolates as suggested. As shown in the figure below, the three distributions are consistent with the observed mixing of egg and chicken isolates (Chicken-vs-Egg includes PADs of all chicken-egg isolates pairs). For the reason we explained above about the perceived utility of PAD analysis, we decided not to include this figure in the manuscript.

References

1. Lemey, P., Rambaut, A., Bedford, T. *et al.* Unifying Viral Genetics and Human Transportation Data to Predict the Global Transmission Dynamics of Human Influenza H3N2. *PloS Pathogens* (2014). <https://doi.org/10.1371/journal.ppat.1003932>

REVIEWERS' COMMENTS

Reviewer #1 (Remarks to the Author):

The authors took all comments seriously and responded to all comments in a satisfactory way. No more questions remain from my side.

Reviewer #2 (Remarks to the Author):

Many thanks for revisions presented in this much improved version of the manuscript. I commend the authors on this work and have no further comments.